# Geographical Indication Characteristics of Aroma and Phenolic Acids of the Changping Strawberry

**DOI:** 10.3390/foods12213889

**Published:** 2023-10-24

**Authors:** Linxia Wu, Xinlu Wang, Jianqiang Hao, Ning Zhu, Meng Wang

**Affiliations:** 1Institute of Quality Standards and Testing Technology of BAAFS, No. 9 Middle Road of Shuguanghuayuan, Haidian District, Beijing 100097, China; wulx@iqstt.cn (L.W.); wangxl@iqstt.cn (X.W.); 2Beijing Center of AGRI-Products Quality and Safety, No. 6 Middle Road of Yumin, Xicheng District, Beijing 100029, China; 13621274015@163.com; 3Beijing Changping Agricultural Technology Extension Station, Science and Technology Center Building, Fuxue Road, Changping District, Beijing 102200, China; lzm629ok@163.com

**Keywords:** aroma, characteristic markers, geographical indication, phenolicacids, strawberry

## Abstract

Strawberry is the most consumed berry fruit worldwide due to its unique aroma and high nutritive value. This fruit is also an important source of phenolic compounds. Changping strawberries are recognized as a national agricultural product of geographical indication (GI) due to their unique flavor. Widely accepted standards for identifying GI strawberries from non-GI strawberries are currently unavailable. This study compared the aroma and phenolic acid composition of GI and non-GI strawberries. Furthermore, the characteristic aroma and phenolic acid markers of GI strawberries were determined. A classification model based on the markers was established using Fisher discriminant analysis (FDA). In this study, six groups of strawberries with variety name of “Hongyan”, including GI strawberries from Changping and non-GI strawberries from Changping, Miyun, Pinggu, Shunyi, and Tongzhou, were collected. A total of 147 volatile substances were discovered using gas chromatography–tandem mass spectrometry. The contents of a few compounds principally responsible for the distinctive aroma in GI strawberries were in the top three of the six groups, providing GI strawberries with a generally pleasant fragrance. OPLS–DA identified isoamyl butyrate and trans-2-octen-1-ol as characteristic markers. Enrichment analysis indicated that beta-oxidation of very long-chain fatty acids, mitochondrial beta-oxidation of very long-chain fatty acids, fatty acid biosynthesis, and butyrate metabolism played critical roles in volatile compound biosynthesis. The total phenolic content was 24.41–36.46 mg/kg of fresh weight. OPLS–DA results revealed that cinnamic acid could be used as a characteristic phenolic acid marker of GI strawberries. Based on the three characteristic markers, FDA was performed on the different groups, which were then divided. The separation of strawberry samples from different origins using the three characteristic markers was found to be feasible. These findings help effectively understand the aroma and phenolic acid composition of strawberries and contribute to the development of strawberries with a pleasant fragrance and health benefits.

## 1. Introduction

Agricultural products of geographical indication (GI) refers to agricultural products that originate from a specific region. Historical, cultural, and environmental variables primarily influence the quality and characteristics of these products [1]. Changping strawberries, a specialty of Changping District, Beijing, was recognized as a GI national agricultural product by the former Ministry of Agriculture in January 2010. Changping District is located in an internationally recognized high-quality strawberry production zone. The unique climatic conditions and contamination-free natural environment contribute to the unique flavor of the strawberries. This fruit also enjoys a high reputation in the surrounding provinces and regions and is exported to Hong Kong, Singapore, and other places [2]. Variety and origin are important factors that influence the quality of strawberries. However, studies on the reasons for the high quality of GI strawberries and the differences between GI and non-GI strawberries are limited.

Strawberries (Fragaria × ananassa [Duchesne ex Weston] Duchesne ex Rozier) are widely grown hybrid species of the genus Fragaria belonging to the family Rosaceae [3]. This berry fruit is popular worldwide due to its high nutritive value, attractive appearance, strong fragrance, and delicious taste; strawberries are also known as the “queen of fruits” [4]. Strawberry fruits currently have more than 350 related volatile components, making them one of the most complex and aroma-rich fruits [5]. Despite accounting for only 0.001–0.01% of the fruit weight, volatile chemicals are crucial in the development of strawberry fruit flavor [6]. Wang et al. [7] conducted comparative research on the fragrance components found in eight strawberry varieties (strains) and detected 46 types of substances, including 23 esters, seven alcohols, six ketones, six aldehydes, and four acids. Parra-Palma et al. [8] identified 48 volatile compounds from four strawberry cultivars, with esters, acids, terpenes, and lactones being the most abundant compounds. Al-Taher and Nemzer [9] identified 29 volatile compounds from the strawberry samples, including terpenes, aldehydes, esters, acids, and alcohols. Similarly, Sheng et al. [10] studied the volatile components in strawberry fruit of 16 varieties and found that their characteristic aroma compounds contained nine esters, six aldehydes, and one alcohol. Chang et al. [11] analyzed the volatile components in three strawberry cultivars with white flesh. The 36 most abundant components were identified. Nogay et al. [12] identified 56 volatile compounds in three strawberry cultivars. Esters, aldehydes, furan derivatives, alcohols, terpenes, acids, and ketones were detected. Fan et al. [13] conducted sensory analyses of strawberry samples grown and harvested over seven years and quantified 113 volatile compounds. Dubrow et al. [14] studied the aroma profiles of strawberries with different liking scores to clarify the relationship between aromas and acceptability and identified nine compounds as predictive compounds. Despite substantial research on the identification of aroma components in strawberries, systematic investigations on the aromatic properties of GI strawberries are scarce. Therefore, widely accepted standards for distinguishing GI strawberries from non-GI strawberries are not available.

In addition to the special aroma, the phenolic chemicals ferulic, p-coumaric, vanillic, caffeic, syringic, and sinapic acids, as well as flavonoids, are abundant in strawberries [15]. Phenolic compounds typically exist in free, conjugated, and bound forms in plant cells, and phenolic compounds in strawberries play a crucial role in their variety of health benefits, including anti-inflammation, anti-cancer, antioxidant, and anti-diabetic properties [4]. Previous research mostly evaluated the differences in the phenolic fingerprint and antioxidant activity of strawberries, raspberries, and blueberries [16], strawberry tree leaves and fruits [17], and different analysis methods [18]. However, limited attention has been paid to the phenolic acid profiles of strawberries with different quality levels.

Clarifying the reasons for the unique flavor of high-quality products (GI strawberries) to a certain extent facilitates the easy identification of products with different qualities in unknown situations. However, there are relatively few well-known characteristic biomarkers to discriminate GI strawberries from non-GI strawberries. Numerous types of biomarkers remain unknown. Therefore, this study intended to introduce the unique characteristics of GI strawberries in terms of aroma and phenolic compounds. High-quality products can be protected by identifying characteristic markers and establishing a discrimination model. The results will provide basic data for strawberry variety improvement and deep processing of functional components.

## 2. Materials and Methods

### 2.1. Materialsand Chemicals

A total of 18 strawberry samples of “Hongyan” strawberries from different districts of Beijing, China, were selected and divided into the following six groups: GI strawberries from Changping (GI) andnon-geographical indication strawberries from Changping (CP), Miyun (MY), Pinggu (PG), Shunyi (SY), and Tongzhou (TZ). Each group contained three samples, which were all kept in ice boxes and brought to the laboratory within 2 h. The materials were weighed, instantly frozen in liquid nitrogen, and stored at −80 °C. Liquid nitrogen was used to finely grind the samples.

Sodium hydroxide (NaOH) and hydrochloric acid (HCl) (analytical grade) were purchased from Beijing Chemical Reagent Company (Beijing, China). Caffeic acid, catechin hydrate, chlorogenic acid, cinnamic acid, 4-dicafleoylquinic acid, 2,3-dihydroxybenzoic acid, 2,5-dihydroxybenzoic acid, 3,4-dihydroxybenzoic acid, 3,5-dihydroxybenzoic acid, ellagic acid, epicatechin, ferulic acid, gallic acid, 4-hydroxybenzoic acid, isoferulic acid, kaempherol, myricetin, neochlorogenic acid, p-coumaric acid, procyanidin B1, procyanidin B2, quercetin, resveratrol, rutin, salicylic acid, sinapic acid, syringic acid, taxifolin, 2,3,4-trihydroxybenzoic acid, and vanillic acid were purchased from Sigma-Aldrich (St. Louis, MO, USA). All the standards used for identification and quantification in this study were of high performance liquid chromatography(HPLC) quality. Methanol, formic acid, and acetonitrile (MS grade) were obtained from Macklin reagent (Shanghai, China). Ultrapure water was obtained from a Milli-Q Element water purification system (Millipore, Bedford, MA, USA).

### 2.2. Aroma Analysis

#### 2.2.1. Headspace Solid-Phase Microextraction (HS–SPME)

The sample powder (500 mg) was immediately transferred to a 20 mL headspace vial (Agilent, Palo Alto, CA, USA) to stop any enzyme reactions. A 120 μm divinylbenzene/carbon wide-range/polydimethylsiloxane-coated fiber was conditioned at 250 °C for 5 min before volatile extraction. The fiber was exposed to the headspace of the vial for 15 min of extraction after a 5 min equilibration in a 60 °C water bath [19]. All measurements were conducted in triplicate.

#### 2.2.2. Gas Chromatography–Tandem Mass Spectrometry Analysis (GC–MS/MS)

Volatile chemicals were analyzed according to the methodology of Yue et al. [20] with minor modifications. After sampling, the injection port of the GC equipment (Model 8890; Agilent) was used to desorb the volatile chemicals of the fiber coatings for 5 min in splitless mode. Volatile compounds were analyzed using an Agilent Model 8890 GC and a 7000D mass spectrometer. Separation was conducted using a DB-5MS (30 m × 0.25 mm × 0.25 m) capillary column (5% phenyl-polymethyl siloxane). The helium carrier gas had a linear velocity of 1.2 mL·min^−1^. The injector and detector were maintained at temperatures of 250 °C and 280 °C, respectively. The oven temperature was set to rise from 40 °C (3.5 min) to 100 °C, 180 °C, and 280 °C with a final holding duration of 5 min, increasing at 10 °C·min^−1^, 7 °C·min^−1^, and 25 °C·min^−1^, respectively. The electron impact ionization mode was utilized, and the mass spectra were recorded at 70 eV. The temperatures of the quadrupole mass detector, ion source, and transfer line were programmed at 150 °C, 230 °C, and 280 °C, respectively. The mass spectra in the *m*/*z* 50–500 amu range were scanned at 1 s intervals. Metabolites were analyzed on the basis of the National Institute of Standards and Technology database. MassHunter was used for integration and calibration.

### 2.3. Phenolic Acid Analysis

#### 2.3.1. Phenolic Acid Extraction

Plants may have three types of phenolic acids: free, conjugated, and bound. In this work, the extraction of phenolic acids followed the method explained by Gao, Ma, Wang, and Feng [21]. Briefly, 20 mL of 80% methanol were used to extract 2 g of strawberry powder, which was then acidified with 1% ascorbic acid. The supernatant was obtained following ultrasonication (30 min, room temperature) and centrifugation (10,000× *g* rpm, 10 min). The supernatant was utilized for free phenolic acids detection. For the analysis of conjugated phenolics, the extraction was repeated and the supernatant was vacuum-evaporated and alkaline-hydrolyzed with 20 mL of NaOH (4 M) in nitrogen blanketing at 35 °C. The hydrolysate was then acidified to pH 2 with HCl (12 M). Hexane (20 mL) was added, and the mixture was agitated for 20 min at room temperature. The resulting hydrolysate was extracted three times with 20 mL of ethyl acetate after eliminating the hexane. At 35 °C, the entire mixture of organic phases was condensed to dryness and then redissolved in 10 mL of 50% methanol/pure water (*v*/*v*).The levels of conjugated phenolic acids were determined by comparing the results of the current and preceding steps. In order to obtain the bound phenolics, the solid residue obtained after the extraction of conjugated phenolics underwent alkaline hydrolyzation and acidification. Oil and other esters were removed after hexane (20 mL) was added and mixed for 20 min. A total of 20 mL of ethyl acetate was used to extract the released phenolics three times. The combined supernatant was vacuum-evaporated to dryness at 35 °C and then redissolved in 10 mL of 50% methanol. Before analysis, the extract from the preceding three stages was finally filtered through 0.22 m PTFE membranes (Pall, Ann Arbor, MI, USA).

#### 2.3.2. Ultrahigh-Performance Liquid Chromatography (UPLC)–MS/MS

According to Gao, Ma, Wang, and Feng [21], strawberry extracts were analyzed by the Waters ACQUITY UPLC system interfaced to a triple quadrupole MS (TQ-S, Waters Micromass, Manchester, UK) and equipped with a photodiode array detector. The separation was conducted using an Acquity HSS C18 (2.1 mm × 150 mm, 1.8 μm)capillary column. A 0.1% formic acid in water (*v*/*v*) gradient system and solvent B (*v*/*v*) were used to elute the samples. Using a flow rate of 0.3 mL·min^−1^, the gradient system was 5% B in A over 30 s, 5–30% B in A over 4.5 min, 30–90% B in A over 4.5 min, 90% B in A was maintained continuously for 30 s, and 90–5% B in A over 30 s. This mixture was then kept for 2.5 min to re-equilibrate. The autosampler was set to 10 °C, while the column was maintained at 45 °C. The injection volume was set at 5 µL. Positive and negative ESI modes were used based on the structural features of phenolic acids. The ESI parameters were as follows: +2.5 kV/1.0 kV for the capillary voltage, 150 °C for the source temperature, 500 °C for the desolvation temperature, 150 L·h^−1^ for the cone gas flow, and 1000 L·h^−1^ for the desolvation gas flow. The detection was conducted in the mode of multiple reaction monitoring. The compounds were quantified in accordance with the calibration curves developed from different compounds in serial dilutions (1–500 ng·mL^−1^) [21].

### 2.4. Statistical Analysis

#### 2.4.1. Principal Component Analysis (PCA)

Unsupervised PCA is a powerful technique to describe significant trends in data. The principal components are created from a data matrix of samples and variables, representing the majority of the original variables’ information [22]. Therefore, only a few principal components could effectively describe a significant variability percentage. In the current study, PCA was used to examine differences between strawberry samples from various groups using MetaboAnalyst (http://www.metaboanalyst.ca, accessed on 11 September 2023). The data were log10-transformed and Pareto scaled before PCA.

#### 2.4.2. Hierarchical Cluster Analysis (HCA)

Cluster analysis is a classified multivariate statistical analysis. Samples are categorized according to the features of individuals, objects, or subjects, wherein individuals in the same and separate categories have the highest homogeneity and heterogeneity, respectively [23]. In the present study, the HCA results of samples and metabolites were displayed as heatmaps with dendrograms, and the Pearson correlation coefficients between samples were computed and displayed as only heatmaps. A color spectrum was used to represent the normalized signal intensities of metabolites (unit variance scaling). The color scale from blue to red reflects the amount of metabolite expression from low to high. HCA was conducted on MetaboAnalyst (http://www.metaboanalyst.ca, accessed on 12 September 2023).

#### 2.4.3. Orthogonal Partial Least Squares Discriminant Analysis (OPLS–DA)

Separating the information from the X matrix into two categories, which are information related to Y and uncorrelated information, OPLS–DA can filter the distinct variables by eliminating the uncorrelated information. To the greatest possible extent, OPLS–DA can effectively simplify the model, improve its interpretation capacity, and maintain its prediction capacity. After an OPLS–DA model between two groups is established, the sample grouping and quantitative information matrices are denoted by Y and X, respectively. The parameters used to evaluate the OPLS–DA model include R2X, R2Y, and Q2Y, wherein R2X and R2Y indicate the interpretation rate of the model to the X and Y matrices, respectively, and Q2Y represents the prediction capability of the model. The model is stable and reliable when the three indicators are close to 1. The model performs better a when Q2Y > 0.5 and excellently when Q2Y > 0.9.

#### 2.4.4. Differential Metabolite Identification

Variable important in projection (VIP > 1) and absolute log2FC (|log2FC| ≥ 1.0) were used in the two-group analysis to identify differential metabolites. Utilizing MetaboAnalyst version 5.0, the VIP values were derived from the OPLS–DA result, which also included score and permutation plots. A permutation test with 200 permutations was conducted to prevent overfitting.

#### 2.4.5. KEGG Annotation and Enrichment Analysis

The identified metabolites were annotated using the KEGG compound database (http://www.kegg.jp/kegg/compound/, accessed on 12 September 2023), and the annotated metabolites were then mapped to the KEGG pathway database (http://www.kegg.jp/kegg/pathway.html, accessed on 12 September 2023) [24]. Mapped pathways with significantly regulated metabolites were fed into metabolite set enrichment analysis, and their significance was determined by the p-values of the hypergeometric test.

#### 2.4.6. Fisher Discriminant Analysis (FDA)

FDA is a commonly used supervised classification method that first establishes a classification model by samples of known labels and then inputs samples of unknown labels into the prediction model [25]. The basic principle of FDA is to project samples from high-dimensional spaces onto low-dimensional spaces; thus, the projected sample data have the minimum intraclass and maximum interclass distances in the new subspace [26]. In the present study, FDA was implemented on SPSS version 21.0 for Windows (SPSS Inc., Chicago, IL, USA).

## 3. Results

### 3.1. Volatile Profile of Strawberry Samples

#### 3.1.1. Comparison of Volatile Compounds and Relative Content

The types and content of volatile compounds have an impact on strawberry aroma. A total of 147 volatile compounds were identified in the analyzed samples (Appendix A). In the analyzed strawberry samples, detected chemical classes included esters (43 compounds), terpenes (21 compounds), aldehydes (21 compounds), alcohols (19 compounds), ketones (16 compounds), acids (10 compounds), lactones (six compounds), furanones (five compounds), and others (six compounds). Several compounds detected in the present study, such as esters (e.g., methyl butyrate, ethyl butyrate, ethyl hexanoate, and methyl hexanoate), furanones (e.g., 2,5-dimethyl-4-methoxy-3(2H)-furanone [DMMF] and its derivative), terpenoids (e.g., linalool and nerolidol), lactones (e.g., γ-decanolactone), acids (e.g., butyric acid), and aldehydes (e.g., trans-2-hexen-1-al), primarily account for the distinct flavor of strawberries [6,27,28,29,30,31].

GI strawberries had different volatile compositions from non-GI strawberries. The variations in abundance of ester, terpenoids, aldehydes, acids, ketones, furanones, lactones, alcohols, and others, are presented in Appendix A. In this study, ester compounds (e.g., methyl isovalerate, isopropyl butyrate, isoamyl butyrate, and methyl-DL-2-methyl butyrate) were the most abundant secondary-metabolite substances in GI strawberries. Among the characteristic aroma substances in strawberries, the level of methyl butyrate was higher in GI strawberries than in non-GI strawberries (*p* < 0.05). An intermediate level of γ-decanolactone was observed in GI strawberries. Terpenes have unique sensory characteristics and are known to have an antibacterial property, which is of particular interest in strawberries [32]. In the present study, GI strawberries exhibited the highest levels of (E)-linalool oxide, cis-linalool oxide, and β-citronellol. Higher contents of linalool and nerolidol were detected in GI strawberries than in non-GI strawberries (*p* < 0.05). Fewer alcohols, aldehydes, and acids were found in GI strawberries than in non-GI strawberries. GI strawberries contained a relatively high level of trans-2-hexen-1-al and caproaldehyde. Among the characteristic volatile substances that have a significant impact on the aroma development of strawberries, DMMF was detected in the current study. Overall, the contents of characteristic aroma components of GI strawberries, which play a dominant role in strawberries, were in the top three of the six groups. For example, the contents of γ-decanolactone, cis-linalool oxide, and (E)-linalool oxide were the highest in GI strawberries; DMMF and butyl acetate were the second highest; ethyl butyrate, methyl butyrate, methyl hexanoate, ethyl hexanoate, trans-2-hexen-1-al, and nerolidol were the third highest. These contents generally provided the GI strawberry with a pleasant fragrance.

#### 3.1.2. Diversity in Volatile Compound Profiles among Different Groups

Figure 1 displays the PCA score plot in the present study. All samples fell within the 95% confidence ellipse. The aroma metabolite composition of the strawberries from the six groups can be distinguished on the PCA load plot. The distribution results revealed considerable differences in the metabolites among the different samples, which could be effectively separated. HCA demonstrated that the species and abundance of metabolites from strawberries showed diversity with the change in origins (Appendix A).

#### 3.1.3. OPLS–DA Models of Volatile Compounds between GI and Other Groups

In the present study, OPLS–DA models were implemented to monitor the variation degree of volatile compounds and determine the differential metabolites between the GI group and other groups. In accordance with the content of volatile compounds and the sample group, the score plot results from the OPLS–DA models between the two groups are presented in Appendix A. The OPLS–DA score plots demonstrated a significant level of discrimination between the GI and non-GI groups. Additionally, excellent model parameters (GI-CP: R2Y = 0.986, Q2 = 0.943; GI-MY: R2Y = 0.997, Q2 = 0.977; GI-PG: R2Y = 0.979, Q2 = 0.960; GI-SY: R2Y = 0.997, Q2 = 0.987; GI-TZ: R2Y = 0.991, Q2 = 0.956) were detected in this study. Thus, the OPLS–DA models were effective and had strong predictive capacity, indicating that OPLS–DA could be utilized to investigate the volatile compound differences between GI and non-GI strawberries.

#### 3.1.4. Determination of Volatile Differential Metabolites and Characteristic Markers

Differential metabolites were identified using VIP (VIP > 1) and absolute log2FC (|log2FC| ≥ 1.0). Overall, 35 differential volatile compounds (32 upregulated and three downregulated) were identified between GI and CP; 40 differential volatile compounds (31 upregulated and nine downregulated) between GI and MY; 31 differential volatile compounds (10 upregulated and 21 downregulated) between GI and PG; 50 differential volatile compounds (39 upregulated and 11 downregulated) between GI and SY; and 18 differential volatile compounds (four upregulated and 14 downregulated) between GI and TZ. Furthermore, the differential volatile compounds were ranked to effectively understand the changes in the major differential volatile compounds between GI and non-GI strawberries (Appendix A). The major differential volatile compounds in GI strawberries were esters and ketones. Finally, the Wenn plot (Figure 2) displayed the relationship between differential volatile compounds in each group. This demonstrated that isoamyl butyrate and trans-2-octen-1-ol could be used as characteristic markers of GI strawberries.

#### 3.1.5. Functional Annotation and Enrichment Analysis of KEGG Differential Metabolite

The annotation results of KEGG metabolites with significant differences were categorized in accordance with the channel types in KEGG [24], and the classification diagram is presented in Appendix A. Except for GI versus CP, no significantly different metabolic pathway was found between the other groups. This result indicated that beta-oxidation of very long-chain fatty acids, mitochondrial beta-oxidation of very long-chain fatty acids, fatty-acid biosynthesis, and butyrate metabolism are crucial in biosynthesis.

### 3.2. Characterization of Phenolic Acids in Strawberry Samples

#### 3.2.1. Quality Analysis of Phenolic Acids

In this study, 30 phenolic compounds were measured in the extracts of strawberry fruits. Chromatograms of 30 types of phenolic acids were presented in Appendix A. Considering the targeted analytes, the total phenolic content was found to be 24.41–36.46 mg/kg fresh weight (FW) in strawberry fruits, demonstrating the dominance of ellagic acid. By contrast, cinnamic, salicylic, and 3,4-dihydroxybenzoic acids showed higher levels in GI strawberries than in non-GI strawberries (*p* < 0.05). However, samples from MY and PG had the highest levels of total phenolic acids, followed by GI (Appendix A). These samples demonstrated that the unique flavor of high-quality GI strawberries might not be attributed to the total phenolic content.

#### 3.2.2. PCA and HCA Results of Total Samples Based on Phenolic Acids

Figure 3 and Appendix A display the PCA score plot and cluster heat map results, respectively, based on phenolic acids. The phenolic acids among the different samples had significant differences and can be effectively separated. The quality of samples from the same group was relatively stable, whereas that from different groups significantly differed. The species and abundance of phenolic acids from strawberries showed diversity with the change in origins.

#### 3.2.3. OPLS–DA Models of Phenolic Acids between GI Group and Other Groups

The OPLS–DA models of phenolic acids between the two groups were established in accordance with the content of metabolites and sample groups. Appendix A displays the score plot output of the OPLS–DA models. The OPLS–DA score plots revealed a high level of discrimination among the groups of samples, revealing a clear division between the GI group and other groups. Excellent model parameters (GI-CP: R2Y = 0.929, Q2 = 0.766; GI-MY: R2Y = 0.979, Q2 = 0.928; GI-PG: R2Y = 0.979, Q2 = 0.902; GI-SY: R2Y = 0.989, Q2 = 0.956; GI-TZ: R2Y = 0.987, Q2 = 0.936) were detected in this study. Thus, the OPLS–DA model can be utilized to investigate the differences between GI and non-GI strawberries based on phenolic acids.

#### 3.2.4. Determination of Phenolic Acids and Characteristic Markers

VIP (VIP > 1) and absolute log2 FC (|log2FC|≥ 1.0) were used to determine differential phenolic acids. Overall, 10 differential phenolic acids (seven upregulated and three downregulated) between GI and CP; 24 differential phenolic acids (seven up-regulated and 17 down-regulated) between GI and MY; 13 differential phenolic acids (three up-regulated and10down-regulated) between GI and PG; 14 differential phenolic acids (seven up-regulated and seven down-regulated) between GI and SY; and 16 differential phenolic acids (10 up-regulated and six down-regulated) between GI and TZ were identified. In addition, the relationship between different phenolic acids in each group was shown through the Wenn plot (Figure 4). It demonstrated that cinnamic acid can be used as a characteristic phenolic acid marker of GI strawberries. It was significantly high in GI strawberries.

### 3.3. Classification of Strawberry Samples from Six Groups Based on Three Characteristic Markers

FDA was performed on the six groups of samples in the present study based on three characteristic markers to monitor the degree of variation of the metabolites (Figure 5). Strawberries were divided into six categories, and GI strawberries were significantly separated from the other groups.

## 4. Discussion

Different types of volatile components present various flavors. Research has shown that esters and lactones endow strawberries with a fruity and floral aroma, whereas furans and some alcohols form a caramel flavor [33]. As one of the numerous aroma substances, γ-decanolactone presents a peach aroma and provides the most contribution to the fruit aroma of strawberries [5,33,34,35,36]. Linalool and nerolidol lend strawberries with a unique rose aroma. Caproaldehyde and trans-2-hexen-1-al give strawberries their distinctive green/vegetative aroma [33,34]. Only 15 odor-active chemicals contribute to the strawberry flavor in commercial strawberry fruits, with esters being the most significant compound [27]. Ester compounds were the most abundant secondary-metabolite substances in GI strawberries in the current study. This finding indicated that GI strawberries had a stronger aroma than non-GI strawberries. Strawberry fruit aromatic compounds are primarily influenced by species, cultivar, agricultural techniques (e.g., soil and climate), breeding programs, maturity, and postharvest environmental conditions [33]. Low amounts of volatile furanone exist in strawberries, which are the main substances that contribute to the sweet aroma; among which, 5-dimethyl-4-hydroxy-3(2H)-furanone (DMHF) and DMMF with caramel and burnt pineapple aromas, respectively, have a significant impact on the aroma development of strawberries [27,37]. Only DMMF was detected in the current study, which could be related to the conversion of DMHF to DMMF under the action of O-methyltransferase and the higher stability of DMMF over DMHF [38]. Fatty acids are the main precursors for the synthesis of aroma compounds in strawberry fruits based on the annotation results of KEGG metabolites. The synthesis process includes α-oxidation, β-oxidation, and lipoxygenase (LOX) oxidation, where β-oxidation and LOX are the main pathways [39].

The findings in the present study agreed with those of Huang et al. [4], who discovered that phenolics in strawberries are primarily non-extractable. The total content of phenolic compounds in strawberries substantially varies based on the cultivar, growing conditions, and maturity, and its estimation may differ depending on the analytical technique [40]. In previous studies, high amounts of phenolics were found in strawberry products, ranging from 25.04 mg/kg FW to 42.06 mg/kg FW [41]. Cervantes et al. [16] quantified the polyphenolic profiles of raw strawberry fruits and found chlorogenic, ellagic, trans-cinnamic, caffeic, gallic, and p-coumaric acids. Previous studies showed that ellagic acid is one of the primary hydrolysable tannins in strawberries, accounting for 81% of the phenolic acids in strawberries [17].

The PCA and HCA results demonstrate a pattern of metabolome separation among groups and indicate the presence of any difference in metabolome within the sample group [42]. The samples from the same group were collected, which showed that the quality of the strawberries was relatively stable from the same group, whereas that of the samples from different groups significantly differed. PCA can effectively extract the main information from high-dimensional data but is insensitive to variables with a small correlation, which can be effectively addressed by PLS–DA. PLS–DA is a multivariable statistical analysis with supervised pattern recognition. Appropriate rotation of the principal components facilitates effective differentiation between group observations [43]. OPLS–DA combines orthogonal signal correction and PLS–DA. Compared with PLS–DA, OPLS–DA can reduce noise irrelevant to classification information and thereby enhance analytical capability, ensure the validity of the models, maximize the differentiation between groups, and help identify different metabolites [44,45]. The VIP value in OPLS–DA is regarded as an effective indicator for assessing the influence intensity and interpretability of the discrimination between groups. The contribution of the variable to the separation is substantial when the VIP value is high. Metabolites with a VIP score greater than one are generally assumed to be differentiated metabolites. Furthermore, fold change (FC) is required to further verify the significance of metabolites between groups. Absolute log2FC (|log2FC|) is frequently used to assess the probability of the difference between groups [46]. The PC1 of the OPLS–DA model for VIP values greater than 1 and |log2FC| ≥ 1.0 was examined to identify potential differential metabolites [47]. Therefore, isoamyl butyrate, trans-2-octen-1-ol, and cinnamic acid were identified as characteristic markers of GI strawberries. Isoamyl butyrate is fruity [48] and was significantly high in GI strawberries. However, trans-2-octen-1-ol display edsoapy and plasticky odors [49] and was undetected in GI strawberries. Cinnamic acid, which has an excellent fragrance preservation function, is typically used as a raw material for flavoring, increasing the fragrance of the main spice [50]. This finding may be related to the intense aroma of GI strawberries. However, further studies are necessary to confirm this relationship. FDA is widely used to distinguish samples from different groups, which indicated the feasibility of separating strawberry samples of different origins using three characteristic markers. In the future, additional samples could be added to validate the model.

## 5. Conclusions

Overall, six groups of strawberry fruits, including GI and non-GI samples, were collected from different districts of Beijing China. The unique flavor of GI strawberries was explained in terms of aroma and phenolic compound aspects to protect the high-quality product (GI strawberries). Characteristic markers were identified, and a discrimination model was established.

A total of 147 volatile compounds were identified by GC–MS/MS. In GI strawberries, ester compounds dominated among secondary metabolites with a relative content of 49.95%. Several compounds that are primarily responsible for strawberry characteristics were detected in the samples. Their contents in GI samples were in the top three of the six groups, generally providing GI strawberries with a pleasant fragrance. This result indicated that GI strawberries had stronger aroma than non-GI strawberries. Isoamyl butyrate and trans-2-octen-1-ol were identified as characteristic markers via OPLS–DA.

In addition, the total phenolic content was 24.41–36.46 mg/kg FW. GI strawberries exhibited higher levels of cinnamic, salicylic, and 3,4-dihydroxybenzoic acids than non-GI strawberries (*p* < 0.05). However, samples from MY and PG had the highest levels of total phenolic acids, followed by GI. This finding demonstrated that the unique flavor of high-quality GI strawberries might not be attributed to total phenolic content. The OPLS–DA results revealed that cinnamic acid can be used as a characteristic phenolic acid marker of GI strawberries.

FDA was performed on the six groups, which were then divided, based on the three characteristic markers. The results showed that distinguishing strawberry samples from different origins was feasible using the three characteristic markers. In the future, additional samples could be added to validate the model. This study provides comprehensive information on the volatile profile and phenolic acid composition of GI and non-GI strawberries.

## Figures and Tables

**Figure 1 foods-12-03889-f001:**
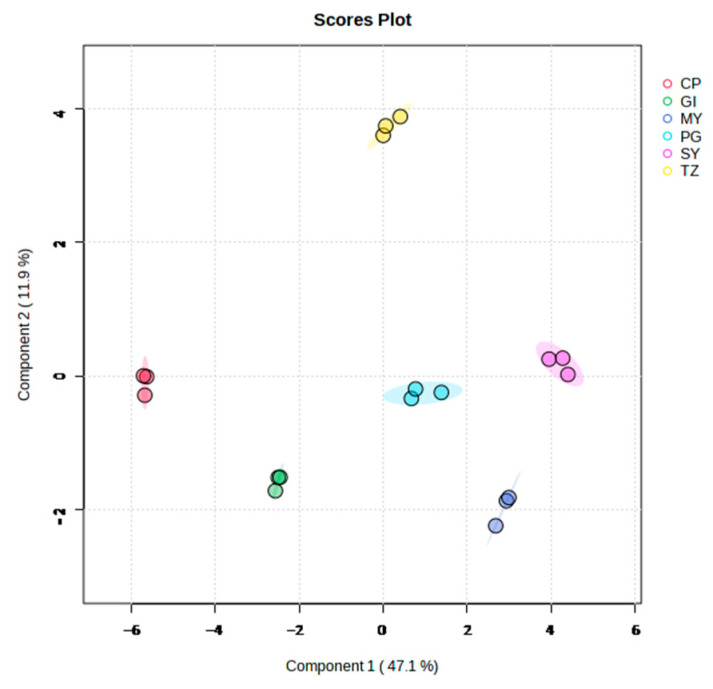
PCA score plot of total samples on the basis of volatile compounds.

**Figure 2 foods-12-03889-f002:**
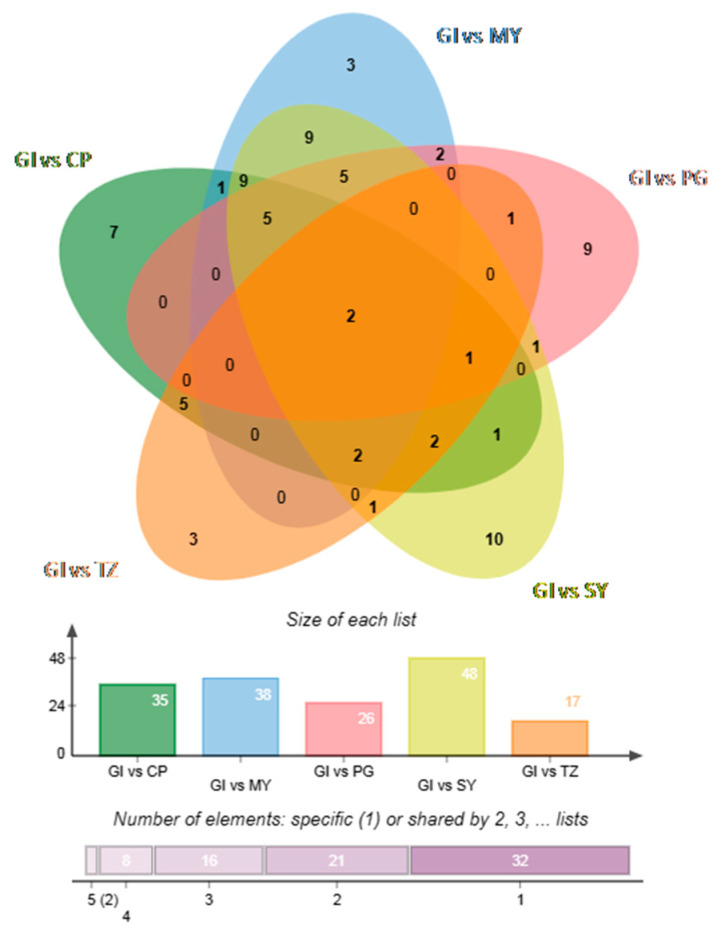
Wenn plot of differential volatile compounds between strawberry samples from the GI group and other groups.

**Figure 3 foods-12-03889-f003:**
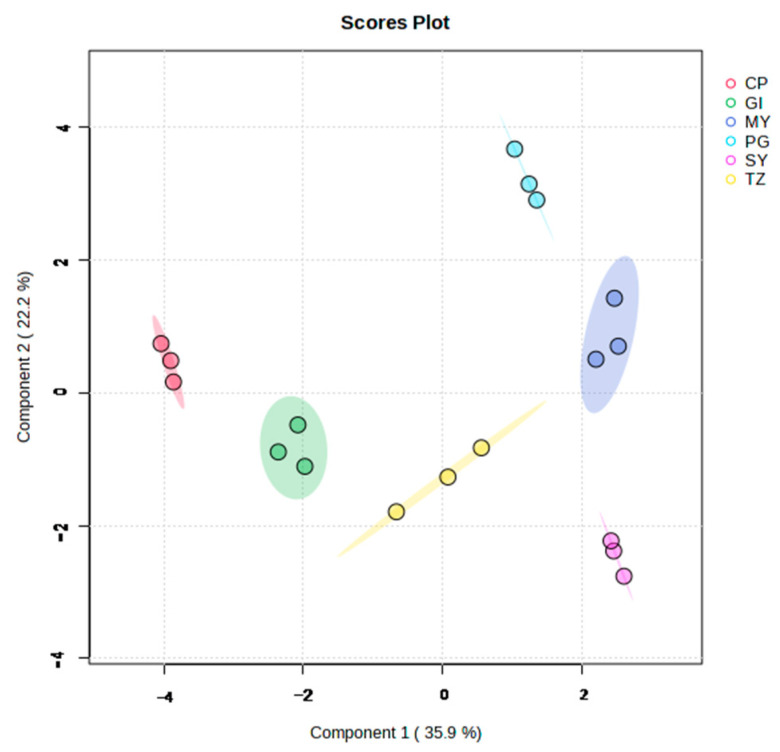
PCA score plot of total samples on the basis of phenolic acids.

**Figure 4 foods-12-03889-f004:**
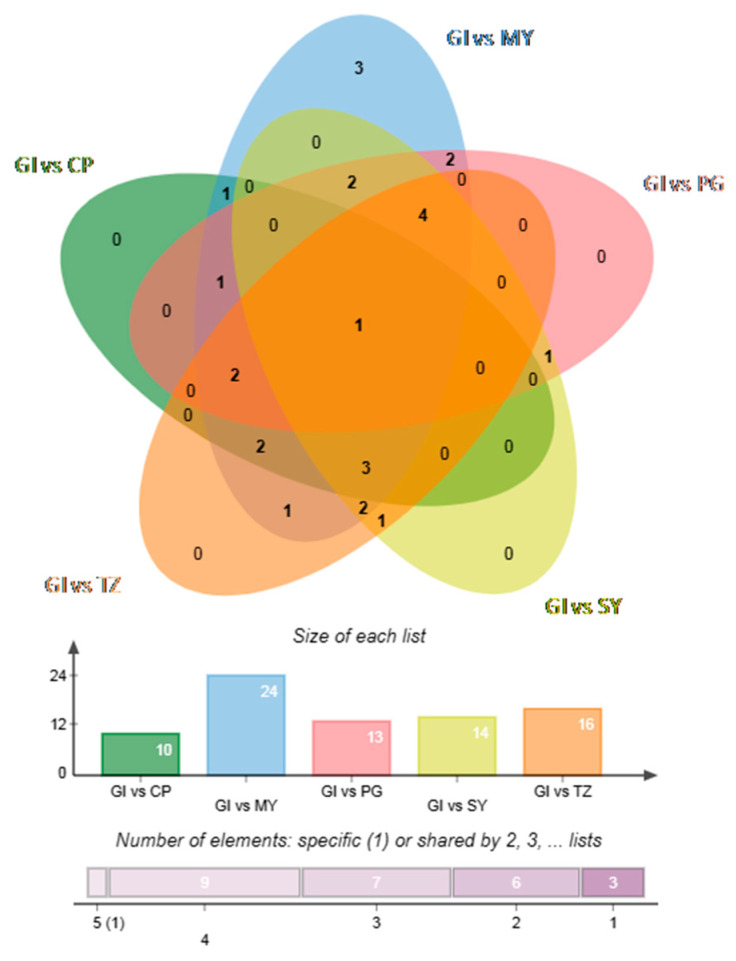
Wenn plot of different phenolic acids between strawberry samples from the GI group and other groups.

**Figure 5 foods-12-03889-f005:**
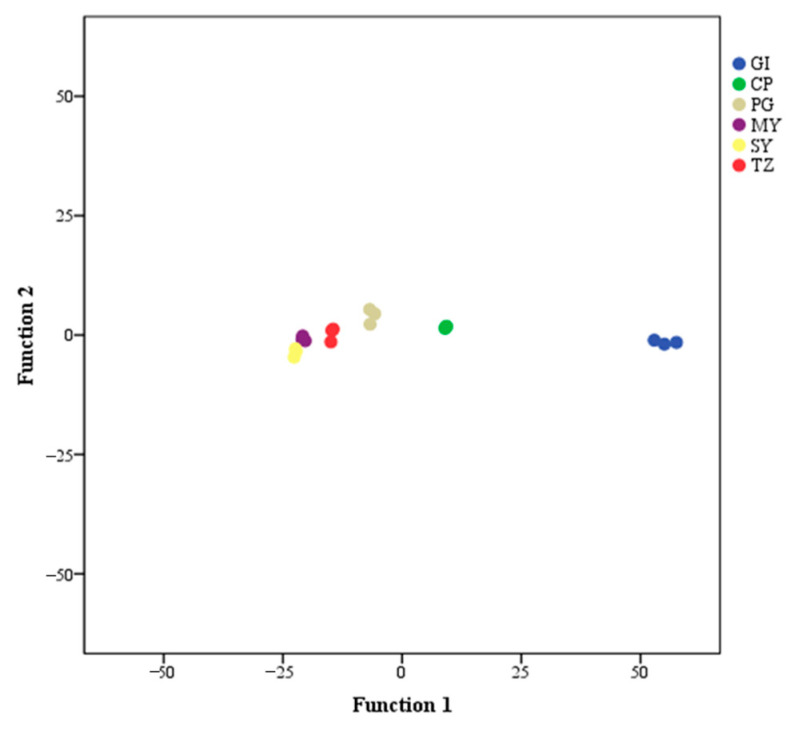
FDA of strawberry samples from six groups by three characteristic markers.

## Data Availability

The data used to support the findings of this study can be made available by the corresponding author upon request.

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
