# Peer review of "Geographical Indication Characteristics of Aroma and Phenolic Acids of the Changping Strawberry"

_foods, 2023, doi:10.3390/foods12213889_

Round 1
Reviewer 1 Report
Comments and Suggestions for Authors
This article is well written on (Geographical indication characteristics of aroma and phenolic acids of Changping strawberry), however here are a few suggestions that will improve the quality of the manuscript if followed by the authors
1. Needs improvement in abstract, more focused on the aim and methodology and conclusion part of the abstract.
2. Add numerical values and original findings of the results at least add those which are important for this study.
3. Add a conclusive line in the end of abstract
4. Mention novelty in this study
5. Explain the statistical analysis in one line in the abstract.
6. Keywords should be written in alphabetical order
7. In the introduction, in cite references should be the latest
8. Add a paragraph in the end of introduction that should describe the importance, reasoning and objectives of the study.
9. Line 77. Material. The authors did not mention where they arranged chemicals for the present study.
10. Authors should provide the complete details of HPLC in methodology section including its detector and column information.
11. Line 86 Headspace solid-phase microextraction (HS–SPME). The authors did not cite any reference for procedure as evidence.
12. Line 93 Gas chromatography–tandem mass spectrometry analysis (GC–MS/MS). The authors did not cite any reference for procedure as evidence.
13. Line 109 please concise the produce of Phenolic acid extraction, as authors had already mentioned reference of the previous study as evidence
14. Line 138 Ultrahigh-performance liquid chromatography (UPLC) –MS/MS. Again The authors did not cite any reference for procedure as evidence.
15. Need Improvement in conclusion
16. Methodology is totally confusing, Authors should go step by step, mention properly, and specially section 2.4, 2.4.1and so on is confusing please revise it.
17. Results are again not in sequence, even not according to methodology.
18. Authors should explain with more clarity
19. Authors should provide proper justification and reasoning in each parameter along with that they should compare with previous studies.
20. Please summarize your conclusion with mean values and provide the in-depth information of this study, reduce its length and add numerical values also declare the best varieties according to their volatile profile.
21. Grammar needs serious attentions; a lot of sentences have no sense.
Comments on the Quality of English Language
Extensive editing of English language required
Author Response
Point 1: Needs improvement in abstract, more focused on the aim and methodology and conclusion part of the abstract.
Response 1: Abstract has been improved by adding the aim and methodology and conclusion part.
Point 2: Add numerical values and original findings of the results at least add those which are important for this study
Response 2: Considering space limitations, numerical values and original findings of the results have been given in the Supplementary materials.
Point 3: Add a conclusive line in the end of abstract.
Response 3: A conclusive line has been added in the end of abstract.
Point 4: Mention novelty in this study.
Response 4: Novelty in this study has been added in abstract and introduction part.
Point 5: Explain the statistical analysis in one line in the abstract.
Response 5: There are six statistical analysis methods in this study. We think it is difficult to explain it in one line. We explained them in 2. Materials and Methods.
Point 6: Keywords should be written in alphabetical order.
Response 6: Keywords have be written in alphabetical order.
Point 7: In the introduction, in cite references should be the latest.
Response 7: We have modified cite references in the introduction.
Point 8: Add a paragraph in the end of introduction that should describe the importance, reasoning and objectives of the study.
Response 8: A paragraph that describes the importance, reasoning and objectives of the study has been added in the end of introduction.
Point 9: Line 77. Material. The authors did not mention where they arranged chemicals for the present study.
Response 9: Chemicals for the present study have been added in Material.
Point 10: Authors should provide the complete details of HPLC in methodology section including its detector and column information.
Response 10: The complete details of HPLC including its detector and column information is provided in Section 2.3.2.
Point 11: Line 86 Headspace solid-phase microextraction (HS–SPME). The authors did not cite any reference for procedure as evidence.
Response 11: Reference for headspace solid-phase microextraction (HS–SPME) procedure has been cited in Section 2.2.1.
Point 12: Line 93 Gas chromatography–tandem mass spectrometry analysis (GC–MS/MS). The authors did not cite any reference for procedure as evidence.
Response 12: Reference for gas chromatography–tandem mass spectrometry analysis (GC–MS/MS) procedure has been cited in Section 2.2.2.
Point 13: Line 109 please concise the produce of Phenolic acid extraction, as authors had already mentioned reference of the previous study as evidence.
Response 13: The produce of phenolic acid extraction has been simplified.
Point 14: Line 138 Ultrahigh-performance liquid chromatography (UPLC) –MS/MS. Again The authors did not cite any reference for procedure as evidence.
Response 14: Reference for ultrahigh-performance liquid chromatography (UPLC) –MS/MS has been cited in Section 2.3.2.
Point 15: Need Improvement in conclusion.
Response 15: Conclusion has been improved.
Point 16: Methodology is totally confusing, Authors should go step by step, mention properly, and specially section 2.4, 2.4.1and so on is confusing please revise it.
Response 16: Because statistical analysis (section 2.4) was utilised to analyze both volatile compound data and phenolic acid data. Therefore, statistical analysis was treated as a new section different with instrumental analysis (section 2.2 and 2.3).
Point 17: Results are again not in sequence, even not according to methodology.
Response 17: Because statistical analysis (section 2.4) was utilised to analyze both volatile compound data and phenolic acid data. Therefore, after data was obtained by instrumental analysis, different statistical analysis methods in section 2.4.1-2.4.6 were applied to analyzed the data.
Point 18: Authors should explain with more clarity.
Response 18: The manuscript has been modified.
Point 19: Authors should provide proper justification and reasoning in each parameter along with that they should compare with previous studies.
Response 19: The justification and reasoning have been provided in Discussion.
Point 20: Please summarize your conclusion with mean values and provide the in-depth information of this study, reduce its length and add numerical values also declare the best varieties according to their volatile profile.
Response 20: Conclusion has been modified.
Point 21: Grammar needs serious attentions; a lot of sentences have no sense.
Response 21: The manuscript has been modified.
Reviewer 2 Report
Comments and Suggestions for Authors
Manuscript ID: foods- 264405
The manuscript entitled “Geographical indication characteristics of aroma and phenolic acids of Changping strawberry” concerns the geographical indication and characteristics of strawberries, based on their aroma composition and phenolic compounds content. The characteristic of aroma and phenolic acids concentrations of strawberries obtained from various districts of Beijing, China have been determined to provide basic data for strawberry variety improvement and deep processing of functional components. Also, principal component analysis has been used to differentiate strawberries samples under different treatments. This study aims to provide the selected plants geographical indication by the determination of characteristic markers. The subject of the work is suitable for the publication in the Foods. However, results presented by authors are not at all sufficient for the original research. Lack of the proper qualitative analysis and presented data are more speculative as a relative. In this reason the paper must be supplied with a particular statistical analysis (validation). All calibration data including retention time, linearity with correlation coefficient of the calibration curves, limits of detection (LOD) and quantification (LOQ), precision as a relative standard deviation (RSD) should be presented. A significant drawback of this study is too small number of analyzed samples.
More details of performed study should be specified, including the report on performance of the proposed method. I recommend the major revision according to specific comments:
· The abstract section lacks important details of source samples including analysed plants’ origin.
· Introduction: the basic information that strawberry is grown hybrid species of the genus Fragaria could be completed.
· Introduction: one of paragraphs (lines: 49-58) no references' numbers provided.
· Materials and methods are usually presented at the end of the manuscript.
· Materials and Methods: incorrect use of units: mL · min-1 and °C · min-1.
· Results: Lack of substantial data in the reviewed manuscript. Example chromatograms should be included.
· Results: Lack of the proper qualitative analysis. The critical evaluation of more important quantitative merits should be completed.
· The references list could be more extensive in the scope of the presented topic.
· It is recommended have to a careful check of the English grammar.
Summary:
The reviewed manuscript has many shortcomings and inaccuracies. The current version of the manuscript needs major revision and requires resubmission.
Comments on the Quality of English Language
It is recommended have to a careful check of the English grammar.
Author Response
Point 1: The abstract section lacks important details of source samples including analysed plants’ origin.
Response 1: The details of samples were added in the abstract section
Point 2: Introduction: the basic information that strawberry is grown hybrid species of the genus Fragaria could be completed.
Response 2: The basic information that strawberry is grown hybrid species of the genus Fragaria has been completed in Introduction.
Point 3: Introduction: one of paragraphs (lines: 49-58) no references' numbers provided.
Response 3: The references' numbers has been provided.
Point 4: Materials and methods are usually presented at the end of the manuscript.
Response 4: Materials and methods are presented according to template.
Point 5: Materials and Methods: incorrect use of units: mL • min-1 and °C • min-1.
Response 5: The units have been modified.
Point 6: Results: Lack of substantial data in the reviewed manuscript. Example chromatograms should be included.
Response 6: Chromatograms of 30 kinds of phenolic acids were presented in Figure S6.
Point 7: Results: Lack of the proper qualitative analysis. The critical evaluation of more important quantitative merits should be completed.
Response 7: We have modified it.
Point 8: The references list could be more extensive in the scope of the presented topic.
Response 8: We have added references.
Point 9: It is recommended have to a careful check of the English grammar.
Response 9: We have modified the manuscript.
Point 10: All calibration data including retention time, linearity with correlation coefficient of the calibration curves, limits of detection (LOD) and quantification (LOQ), precision as a relative standard deviation (RSD) should be presented. A significant drawback of this study is too small number of analyzed samples.
Response 10: Calibration data was basically similar to our previous published article "Gao, Y.; Ma, S.; Wang, M.; Feng, X.Y. Characterization of free, conjugated, and bound phenolic acids in seven commonly consumed vegetables. Molecules 2017, 22". To prevent plagiarism, it was not presented in this study. Each sample is a mixture of strawberries from 20 orchards, and the samples are definitely representative. We will improve it in the future studies.
Round 2
Reviewer 1 Report
Comments and Suggestions for Authors
After checking the grammar and citations, As i still feel some mistakes in language and citations.
Overall authors have revised their manuscript in good manners.
Comments on the Quality of English Language
Some grammatical errors are still there which should be addressed by using any software.
Author Response
Point 1: After checking the grammar and citations, As i still feel some mistakes in language and citations.
Response 1: English language has been edited. Proof of review was attached as follows.
Reviewer 2 Report
Comments and Suggestions for Authors
Manuscript ID: foods- 264405_rev2
The revised manuscript in some parts has been improved according to my comments. My questions have been answered, but not at all. The answers are incomplete and short as well. Still needs improvement in abstract. The authors did not cite appropriate references in Introduction (References should be given in brackets [??]).
Summary:
The current version of the manuscript needs minor revision
Comments on the Quality of English Language
Moderate editing of English language required
Author Response
Point 1: Still needs improvement in abstract.
Response 1: Abstract has been improved.
Point 2: The authors did not cite appropriate references in Introduction (References should be given in brackets [??]).
Response 2: We have modified the reference.
Point 3: Moderate editing of English language required
Response 3: English language has been edited. Proof of review was attached as follows.